# *"It would really support the wider harm reduction agenda across the board"*: A qualitative study of the potential impacts of drug checking service delivery in Scotland

Danilo Falzon[1]*, Tessa Parkes[1], Hannah Carver[1], Wendy Masterton[1], Bruce Wallace[2], Vicki Craik[3], Fiona Measham[4,5], Harry Sumnall[6], Rosalind Gittins[7], Carole Hunter[8], Kira Watson[9], John D. Mooney[10], Elizabeth V. Aston[11]

1 Salvation Army Centre for Addiction Services and Research, Faculty of Social Sciences, University of Stirling, Stirling, United Kingdom, 2 Canadian Institute for Substance Use Research, University of Victoria, Victoria, Canada, 3 Public Health Scotland, Glasgow, United Kingdom, 4 Department of Sociology, Social Policy and Criminology, University of Liverpool, Liverpool, United Kingdom, 5 The Loop, Registered Charity, Manchester, United Kingdom, 6 Public Health Institute, Liverpool John Moores University, Liverpool, United Kingdom, 7 Via, London, United Kingdom, 8 Alcohol and Drug Services, NHS Greater Glasgow and Clyde, Glasgow, United Kingdom, 9 Scottish Ambulance Service, Edinburgh, United Kingdom, 10 Public Health Directorate, NHS Grampian, Aberdeen, United Kingdom, 11 School of Applied Sciences, Edinburgh Napier University, Edinburgh, United Kingdom

* d.c.falzon@stir.ac.uk

**Data Availability Statement:** We have significant ethical concerns about making the transcripts available in a public repository. The small number

## Abstract

Drug checking services (DCS) enable individuals to voluntarily submit a small amount of a substance for analysis, providing information about the content of the substance along with tailored harm reduction support and advice. There is some evidence suggesting that DCS may lead to behaviour and system change, with impacts for people who use drugs, staff and services, and public health structures. The evidence base is still relatively nascent, however, and several evidence gaps persist. This paper reports on qualitative interviews with forty-three participants across three Scottish cities where the implementation of community-based DCS is being planned. Participants were drawn from three groups: professional participants; people with experience of drug use; and affected family members. Findings focus on perceived harm reduction impacts of DCS delivery in Scotland, with participants highlighting the potential for drug checking to impact a number of key groups including: individual service users; harm reduction services and staff; drug market monitoring structures and networks; and wider groups of people who use and sell drugs, in shaping their interactions with the drug market. Whilst continued evaluation of individual health behaviour outcomes is crucial to building the evidence base for DCS, the findings highlight the importance of extending evaluation beyond these outcomes. This would include evaluation of processes such as: information sharing across a range of parties; engagement with harm reduction and treatment services; knowledge building; and increased drug literacy. These broader dynamics may be particularly important for evaluations of community-based DCS serving individuals at higher-risk, given the complex relationship between information provision and health behaviour change which may be mediated by mental and physical health, stigma,

of participants in each city makes it impossible to guarantee that participants cannot be identified through the transcripts. We also did not ask for our participants' consent to make the transcripts available publicly. This was a choice based on the sensitive nature of the topics covered in the interviews including criminalised activity. Other researchers who want to replicate the study can contact the corresponding author and/or the NICR ethics committee at University of Stirling (ethics@stir.ac.uk)

**Funding:** This research was funded by Drug Deaths Taskforce/Corra Foundation, grant number 20/5304. Corra foundation: https://www.corra.scot/ Hannah Carver (grant number 20/5304) Tessa Parkes (grant number 20/5304.) Wendy Masterton (grant number 20/5304.) Danilo Falzon (grant number 20/5304.) Elizabeth V. Aston (grant number 20/5304) Vicki Craik (grant number 20/5304) Fiona Measham (grant number 20/5304) Harry Sumnall (grant number 20/5304) Bruce Wallace (grant number 20/5304) The funders played no role in the study design, data collection and analysis, decision to publish, or preparation of the manuscript.

**Competing interests:** The authors declare no conflict of interest. The funders had no role in the design of the study; in the collection, analyses, or interpretation of data; in the writing of the manuscript; or in the decision to publish the results.

criminalisation and the risk environment. This paper is of international relevance and adds to existing literature on the potential impact of DCS on individuals, organisations, and public health structures.

## Introduction

Global drug markets are increasingly complex and pose risk of drug-related harms to people who use drugs [1–7], although risk and volatility can vary substantially across regions. This dynamic is, in part, driven by the proliferation in number, type, and availability of 'novel' drugs [3, 8]. In the European Union, for example, 880 types of novel psychoactive substances (NPS) are currently being monitored, including 224 variations of cannabinoids, 162 cathinones, 73 opioids, and 33 benzodiazepines [3]. In North America, the emergence of fentanyl (s) in the opioid market has precipitated the 'fourth wave' of an overdose crisis [1, 5]. In some areas of Canada, up to 96% of expected opioid samples contain fentanyl or its analogues [9]. Adulteration of opioids with benzodiazepines ('benzo-dope') and xylazine ('tranq-dope') is increasingly prevalent in the Canadian drug supply [10]. Such dynamics suggest that supply-side factors (complex, unregulated drug markets) are an increasingly significant driver of the overdose and poisoning risks faced by people using drugs. In response, policy makers, public health officials, researchers, and activists, including people who use drugs, have called for the implementation of interventions which address these risks [11–16].

### DCS implementation in a Scottish context

There has been an increase in drug-related deaths in Scotland and rates are currently amongst the highest globally per head of population [17]. Increased use and availability of non-prescribed, 'novel' benzodiazepines, often used in combination with alcohol, opioids, and other substances, are a significant contributing factor. Novel benzodiazepines (sometimes called 'street benzos' e.g., etizolam) can be more potent than prescribed benzodiazepines, but are often illicitly "prepared and branded" to mimic the appearance of licensed benzodiazepines such as diazepam [6, p.3]. The 'street benzo' market in Scotland continues to diversify and evolve, as emerging novel benzodiazepines are frequently detected [18, 19]. There has also been a recent increase in the detection of 2-Benzyl benzimidazole ('nitazene') and other novel opioids in Scotland, raising concerns around the potential for further increases in drug-related deaths. Four nitazene analogues have been detected in Scotland (N-pyrrolidino-etonitazene, metonitazene, protonitazene and isotonitazene), largely as counterfeit oxycodone tablets in the community and combined with synthetic cannabinoids infused on blotter paper in prisons [20]. Many nitazenes are of a similar, or greater, potency to fentanyl. For example, N-pyrrolidino-etonitazene is approximately 20 times more potent than fentanyl [20]. Given Scotland's already complex drug market, and high levels of drug-related deaths, there is a need for urgent harm reduction solutions [21].

Drug checking services (DCS) are one intervention currently being explored in Scotland as part of a suite of harm reduction responses to drug-related harm [22–24]. DCS enable people who use drugs to submit a substance for analytical testing, providing information about the analysed substance as part of a tailored harm reduction consultation [25]. In the UK, the Welsh Emerging Drugs and Identification of Novel Substances project (WEDINOS) (a postal drug testing service which does not offer individually tailored consultations) analyses samples submitted by post from throughout UK, including Scotland, and has operated since 2009 [26].

In 2016, drugs charity The Loop introduced the first DCS at festivals in the UK [27], and are planning to open a regular, community drug checking facility in the city of Bristol, funded by the city council [28]. There are currently no DCS operating in Scotland, but delivery is being planned in Aberdeen, Dundee, and Glasgow.

## Existing evidence on the harm reduction impact of DCS

While DCS have primarily been framed as an intervention providing harm reduction support to "recreational drug users not generally seen at other services" [29 p.2], service delivery models and target demographics are diversifying as DCS proliferate globally [30]. For example, community-based DCS in North America have been established in low-threshold harm reduction settings, such as injecting equipment provision and overdose prevention sites, and aim to engage those at highest risk of experiencing drug harms [31–33]. Diversification of service delivery poses questions concerning the impact of DCS and how to best evaluate outcomes.

Although DCS are increasingly being implemented as a response to rising drug overdose rates across a number of countries, the evidence base is relatively sparse. The majority of studies thus far have focused on processes, trend monitoring, comparison of analytical methods, and service user demographics [9, 32, 34–45]. However, recent research has begun to assess individual level behavioural outcomes, i.e., the impact of DCS engagement on the drug use patterns and behaviours of individuals [25, 27, 42, 43, 46–50]. Most evidence of individual behavioural outcomes is from festival-based DCS catering to so-called 'recreational' drug users, although there have been a small number of evaluations which have included more marginalised groups at higher risk of experiencing drug-related harms [46, 47, 51]. Whilst not DCS in the strict sense [30], a number of evaluations of fentanyl test strip distribution interventions have found that those that have received the intervention report intent to adopt of a number of harm reduction practices [52–54].

Qualitative studies of community-based DCS have called attention to the potential of these services to foster increased drug literacy and engagement in harm reduction, and to the dissemination of drug information through social networks [55–59]. Such processes may be particularly important for capturing the impact of DCS more broadly, as a number of studies have found that a significant proportion of people who access DCS do so after having already consumed the substance submitted for testing [46, 47, 55]. Further, studies report that individuals commonly access DCS to test substances on behalf of others as well as themselves [40], adding nuance to the hypothesised relationship between information provision and subsequent adoption of safer drug use practices.

Beyond outcomes related to individual drug use behaviour, it has been argued that DCS may have an impact at various 'levels' of influence [27, 58]. The strongest evidence for such impacts relates to drug checking's potential to enhance systemic capacity for drug market monitoring [32, 34, 35, 39, 41–45]. Indeed, there are several examples of drug checking detecting and monitoring substances of concern and emerging market trends, helping to build public health intelligence and inform targeted responses [7, 39, 42, 45, 47, 60]. DCS can feed into drug market monitoring structures at the local, national, and international level [47, 61], with potential for more timely and targeted information than that provided through testing of police seizures or toxicological investigation.

A emerging strand of DCS literature also suggests the potential for drug checking to work as a grassroots form of supply-side drug market regulation, by providing people who use and sell drugs with greater autonomy and knowledge to inform decision making when buying and selling drugs [40, 56, 59, 62, 63]. As noted, multiple studies have found that individuals intend to share results more widely with their social network, suggesting a potential impact for drug

checking extending beyond the individual accessing the service [25, 27, 43, 47, 48, 55]. Sources have also reported that service users may return to their supplier to inform them of results or return products which were other-than-expected [25, 63].

There is also evidence that those who sell drugs may use DCS to gain information about their products. For example, 12% of those accessing a Canadian DCS reported doing so for the purposes of selling drugs [40]. Qualitative studies in Canada have observed use of DCS amongst those who sell drugs, who report that DCS can inform practices such as: returning products or information to suppliers further up the chain; cutting batches where they are stronger than expected; informing clients of drug checking results; and being wary of selling high strength products to customers with a low tolerance [62, 63]. It has been argued that this may point to the potential for drug checking to facilitate an upstream form of drug market/ supply intervention, where more accurate information about the contents of drugs are disseminated into the wider community [58, 59].

Despite the emerging evidence base described above, there remain several gaps. For example, there is currently minimal evidence related to whether DCS have impacted on population-level indicators of harm such as drug-related hospitalisations, uptake of wider drug treatment, and drug-related deaths [27, 64]. This paper contributes to addressing such evidence gaps, reporting on data from a larger research project exploring implementation and feasibility considerations for DCS in Scotland. The paper explores participant perceptions of the potential harm reduction impacts of drug checking delivery, aiming to inform service delivery and future evaluation.

### Theory informing the study

The paper draws on Wallace et al.'s socio-ecological model of drug checking (2021), as well as the application of micro, meso and macro spheres of influence by Measham (2019, 2020), to inform analysis and reporting [27, 47, 58]. Wallace and colleagues discuss that drug checking may have impacts at various levels including: on individuals accessing the service; on the wider community; on the drug market in terms of increasing accountability and available information; and at the level of public policy by mediating policies around substance use including safer supply and mitigating harms from criminalisation [58]. A socio-ecological perspective "focuses on how social, political, and economic factors as well as features of the physical environment interact with personal characteristics to determine health" [65, p.126-217]. As noted, much of the existing evidence base around the impacts of DCS has focused on individual level changes to drug use behaviour amongst people who use drugs recreationally. However, an exclusive focus on individual level outcomes may neglect the broader impact of drug checking provision across multiple levels of influence.

### Materials and methods

This paper draws on findings from interviews with key stakeholders across three cities where DCS implementation is being planned in Scotland. Participants were drawn from three groups: professional participants (including police, National Health Service (NHS), and third sector/not for profit staff); people with experience of drug use (either currently or in the last 12 months); and family members of people with experience of drug use. These participant groups were selected based on their varied expertise and experience relevant to informing drug checking implementation. Ethical approval for the study was granted by University of Stirling's NHS, Invasive, and Clinical Research (NICR) panel (paper 0562; March 2021). NHS Research and Development approval was granted from each of the three NHS boards involved (for interviews with NHS staff only). Participant recruitment for the study was undertaken between 25th March-29th September 2021.

Eligibility for the study, and recruitment, consent, and debrief processes, have been described in detail elsewhere [66, 67]. All interviews were conducted by telephone by DF/WM, lasted an average of 51 minutes (range: 14–87 minutes), and were audio recorded. Interview schedules were designed to ask about a wide range of issues relating to drug checking, including: challenges around implementation; the parties who should be involved in implementation; suitable locations and service delivery models; barriers and facilitators to engagement amongst different groups of people who use drugs; policing and community challenges to delivery [23]; general perceptions of drug checking; and the potential impacts of delivering DCS. Participant discussion of the potential harm reduction impact of DCS is the focus of the current paper. Other papers from the study explore topics such as policing of DCS [23], and key implementation and service design considerations [66, 67].

Data were analysed using thematic analysis [68]. Initial coding was conducted using an inductive approach, with a selection of transcripts (n = 16) from a mix of participant groups coded by DF to create an initial coding framework. This framework was checked by the wider team (WM, TP, HC), with themes discussed and alterations made. Once the initial framework was agreed upon, two researchers (DF, WM) coded the remaining transcripts using the framework, meeting frequently to discuss identified themes and make any relevant alterations. Once general coding of all transcripts was conducted, discussions amongst the research team identified the potential impacts of drug checking as a major theme in the data. The data relevant to this topic were then refined and further coded, with frequent discussion and input across the research team. The final stages of coding for this paper drew upon Wallace et al.'s socio-ecological model of drug checking [58] and the work of Measham [27, 47]. These models were used deductively to inform the sorting of codes about the impact of drug checking into various 'levels' of influence. The models were selected after discussion identified that participants did not only focus on drug checking as a means of altering individual drug use behaviours but instead placed emphasis on nuanced potential outcomes across a range of levels and groups, including harm reduction staff, wider organisations, the drug market, and communication networks. The various levels presented in the results do not correspond perfectly with the source materials; instead the frameworks were used as a guide for the deductive coding process.

## Results

Forty-three interviews were conducted across three stakeholder groups. Twenty-seven participants in professional roles were interviewed, including 10 working in Police Scotland, nine in NHS roles, and eight across the third (not for profit) sector. Eleven participants with experience of drug use and five family members were interviewed. Professional participants worked across a range of roles including frontline, managerial, and strategic decision-making positions. Participants with experience of drug use reported current or recent use of drugs including heroin, powder or crack cocaine, and novel benzodiazepines. Family member participants were all parents of an adult currently or recently using drugs dependently. Detailed participant demographics have been reported elsewhere [67].

Themes are reported in four overarching 'levels' of potential impact: individual-level impacts; meso-level impacts; market-level impacts; and macro-level impacts (see Table 1).

### Individual-level impacts

Participants across all groups described DCS as having an impact at the 'individual' level (i.e., for the person accessing the service). They described the benefits of people having access to greater information about the composition of drugs. It was felt that this information could

**Table 1. Themes reported.**

| |
| --- |
| **Individual-level impacts** |
| *Theme 1*: Increased availability of information about drug composition |
| *Theme 2*: Impacts on patterns of drug use |
| Meso-level impacts |
| *Theme 3*: Drug checking and engagement in wider harm reduction |
| *Theme 4*: Drug market monitoring and communication |
| Market-level impacts |
| *Theme 5*: Increased pressure from consumers |
| *Theme 6*: Use of drug checking among people who sell drugs |
| Macro-level impacts |
| *Theme 7*: Reductions in macro-level indicators of drug-related harm |

lead to changes in drug use towards safer practices, but there were a number of barriers to this translation process, centering on people's risk environments, their mental health, and high levels of drug use (often using multiple drugs concurrently). Given such challenges, enacting safer drug use practices was often described as a complex and non-linear process. For this reason, the relationship between information provision and behaviour change was seen as part of a broader process of increasing drug/risk literacy at an individual and community level.

**Increased availability of information about drug composition.** Participants described a primary role for DCS in increasing the availability of information about the composition of drugs, often described as an underpinning rationale of drug checking, as expressed by one participant: "*because, fundamentally, you want to know what you are about to ingest*" (Professional participant 26, third sector). The provision of reliable information was described as increasingly important due to the changing nature of the drug market. One participant noted that: "*there was always dodgy drugs around years ago with maybe a higher concentration of things but yeah. . . nothing like it is now*" (Professional participant 25, NHS). The proliferation of novel benzodiazepines, growth in purchases of drugs from the dark-web, increased complexity of drug samples, and a dynamic and volatile supply, were described as contributing to the need for provision of reliable information:

> We know that people buy online, drugs that they are misusing, a lot of, benzodiazepines, pain killers, gabapentinoid, hypnotics, we know that these are provisioned all over the world through internet sites and so again I think it's a service for people whose drug use is more around those kinds of products to know what they are actually getting and understand how they might interact and the harms that they might get and particularly things like risk of overdose from them. (Professional participant 10, NHS)

Participants with experience of drug use described a number of benefits to drug checking, including: information about the contents and potency of a drug; increased awareness of the implications of ingesting a drug for their mental and physical health; and the opportunity to find out whether they had been "*ripped off*" (PWEDU participant 2). Knowing whether the contents of a drug had been mis-sold was often described as important. As participants noted, people want to know "*if [a drug] is actually going to give [them] a hit*" (PWEDU participant 11) and would access DCS to ensure that what they had bought was "*what it said on the can*" (PWEDU participant 4). It was further noted that people may access drug checking after having used a drug if it had not had the expected intoxicating effect, they had experienced negative effects, or if the drug had been associated with an overdose within their social circle.

The provision of information from DCS was seen as having the potential to increase people's sense of control around what they are ingesting, albeit often with significant caveats in relation to an individual's circumstances and risk environment:

*People who are using street drugs know that they don't have control. If you give them control by checking their drugs, that is a positive thing. So, I know that if I was feeling positive and in control then that's going to boost my self-esteem, it's a natural progression from lack of control.* (Family member participant 1)

Several participants noted a lack of control around what they were taking. As described by one participant: "*I don't even know what I'm taking, I just know I'm taking it*" (PWEDU participant 1). DCS were seen as increasing people's sense of control by providing them with information, for example: "*reassurance that what they are away to take isn't ten times stronger than [usual]*" (Professional participant 17, police).

**Impacts on patterns of drug use.**   Participants felt that information provision could lead to changes in patterns of drug use in a number of ways, including: taking less and measuring dosage according to the information provided; mixing substances less; changing the route of administration (e.g., from injecting to smoking); disposing of a substance in the event of an unexpected result; and returning to a supplier or seeking a new supplier. Participants tended to place emphasis on potential changes to patterns of drug use, rather than on disposal of substances in the event of an unexpected or concerning result. This was due to DCS in Scotland being conceptualised primarily as services aimed at engaging those at highest risk of experiencing drug-related harm. One participant described their belief that those at higher risk would be less likely to dispose of substances of concern:

*Once people are fully informed of what's in it, it won't change the fact that they are using the drug or not. . . I know that people talk about 'oh well people might safely dispose of the drugs'. The problem is we are looking at that through the lens of people attending festivals and stuff like that right. That is a totally different client group who might make that decision to dispose of it because it looks too strong. This group aren't going to do that.* (Professional participant 4, third sector)

A range of barriers were discussed relating to the translation of information provision into the adoption of harm reduction practices. It was noted that more marginalised individuals often navigate complex risk environments, experience poor mental and physical health, and may engage in patterns of heavy drug use and poly-drug use. One participant described people often taking several dozen novel benzodiazepines at once: "*well I've heard of people at chemists talking about 'oh they've took like seventy Valium that morning'*" (PWEDU participant 11). It was noted that people may not always be in the right space to engage with DCS, or to take on board the harm reduction advice provided. For example, one participant described a sense of ambivalence to risk:

*We know the risks. We know that they are not proper pharmaceutical Valium, but we will take them anyway because. . . we've low self-worth and all that sorts of thing. So, we are really looking for a quick fix, but a really dangerous quick fix, because we are used to living in danger. (*PWEDU participant 5)

A further perceived concern was that people may use DCS to seek out potent drugs. Indeed, some participants noted that they would seek to buy the most potent drugs available. As

expressed by one participant: "*if you find out somebody has overdosed most people will say 'where did you get that from? I want some'*" (PWEDU participant 6). However, this was also described as having harm reduction value for some individuals, by allowing them to alter their patterns of use or mode of administration:

> *If I knew it was 43% pure and I knew I was going to get the same out of smoking it as injecting it. . . it's because the drugs are so weak here, I don't feel that I can smoke it. It costs too much money. . . Yeah, I would probably smoke it.* (PWEDU participant 6)

Owing to the myriad of complex factors mediating between information provision and safer patterns of drug use, participants tended to emphasise the more general value of DCS in contributing to increased drug literacy amongst service users. For example, it was felt that DCS may: help people better understand the differences in potential potency between different types of novel benzodiazepines; increase overdose awareness; improve knowledge of different drug half-lives; increase awareness of the effects of different types of poly-drug use; inform people of the interactions between drugs and different types of medications; and encourage sharing of accurate information through social networks. Changes in patterns of drug use were described as complex processes contingent on a number of factors. As expressed by one participant: "*human beings are complex and there is a whole range of responses [to information provision]*" (Professional participant 15, third sector). It was noted that people may have 'good days' and 'bad days', and that, while someone may not be able to able to enact harm reduction behaviors on a bad day, they could carry this increased understanding with them and enact it at a later date:

> *I don't know if it would prevent them doing it that day, but it might make them think differently about doing it another day, depending on where they are at.* (Family member participant 3)

Despite this perception of drug checking as a broader process of increasing drug literacy, participants noted significant challenges in relation to encouraging shifts in "*thought process* [es] *and habit*[s]", particularly in relation to those facing intersecting challenges around marginalisation, discrimination, and structural vulnerability:

> *I presume it's the same across the UK, or certainly in Scotland. 'Oh, those are white pills, I bet they are like diazepam, I will just take that', and not thinking too much more about it. How do you change that kind of thought process and that habit?* (Professional participant 23, NHS)

Contributing to this conceptualisation of DCS as an intervention that could increase drug-related knowledge more generally was the perception that people, particularly those who take drugs daily, may utilise the service having already consumed some of the substances submitted for testing, rather than accessing DCS prior to use. Engaging in drug checking post-consumption was still perceived as useful by most participants for several reasons. Firstly, and as discussed below, it provides those using the service with the opportunity to gain information about their supplier and potentially change source. Secondly, participants noted that drug checking could present opportunities for more tailored and detailed conversations about drug use:

> *If someone has come in and then said 'these are rubbish, can I get these tested? I've taken six of them and I'm getting nothing?'. . . because that onset hasn't kicked in, then you are testing*

*them saying 'no these are very potent, it's etizolam, these are going to kick in and what else have you had today?'. 'Oh, I've picked up my methadone. And I had a charge this morning and I've also got some gabapentin on me'. . . 'Right okay', then we can kind of lay it out.* (Professional participant 2, third sector)

## Meso-level impacts

Participants described a number of potential meso-level impacts which DCS may have. 'Meso-level' can be broadly understood as a heuristic for studying the norms, interactions, and processes of social groups, communities and institutions in finer detail than through a macro-structural lens [69]. The meso-lens extends the unit of analysis beyond the individual by focusing on groups and sets of interactions, processes, and beliefs. Here the focus is on the potential impact of DCS on harm reduction services and staff, and on local and national drug market monitoring structures and processes.

**Drug checking and engagement in wider harm reduction.** Participants did not generally perceive of DCS as standalone services, but rather as interventions integrated into existing sites that offered a range of harm reduction supports:

*You will have wider harm reduction conversations around safer injecting, you'll have wider access to naloxone. You will create wider access to the needle exchange.* (Professional participant 5, third sector)

Many participants noted that drug checking may both encourage engagement amongst those not currently accessing harm reduction support and strengthen engagement among a site's existing service users. Drug checking was therefore viewed as having the potential to increase engagement in wider harm reduction support among a broad range of individuals, including: younger people with shorter histories of drug use and people using drugs on a 'recreational' basis; people using frequently who were hiding their use from family and friends; and people using drug-types which were currently under-represented in service demographics such as stimulants. One participant noted that these groups: *"are just not known to services. . . so I think if you could just extend that reach into those different populations that would be one outcome"* (Professional participant 10, NHS). Engaging younger people with shorter drug use histories was seen as having preventative potential, by providing support to individuals before they experienced significant harm from drug use:

*Especially for the younger generation as they are growing up. . . Like I know that some of them have seen their parents use this and do that and ken [you know], maybe they will be a bit wiser to find out what's in it.* (PWEDU participant 10)

Participants also discussed the potential for DCS to strengthen engagement and trust in the service and its staff amongst existing service users. It was noted that drug checking was a means of signifying that a service cared about individuals and wanted to support them to stay safe. The discussion of one participant was instructive of this point, as they noted: *"[drug checking is] that other kind of foot in the door where someone goes 'oh alright okay they actually understand and they have an idea of how I'm feeling and why I'm doing this'"* (Professional participant 2, third sector). Particularly in relation to more marginalised individuals facing multiple, complex challenges, participants felt that drug checking should be conceived of as just one part of a broader project to provide comprehensive, wrap around care and support within a low-threshold harm reduction environment: *"I think it [drug checking] would really support the wider harm reduction agenda across the board"* (Professional participant 5, third sector).

**Drug market monitoring and communication.** Participants felt that DCS could increase systemic capacity for drug market monitoring and subsequent information sharing, with several harm reduction benefits. In this sense, the benefits of drug checking were seen as extending well beyond individual service users, with benefits for both frontline services and people who use drugs who were not accessing DCS: *"it's really about education. Education for our partners, our staff, and our service users"* (Professional participant 3, third sector). It was felt that drug checking could help build a more accurate picture of local and national drug markets. Participants noted that current drug market information, commonly provided through toxicology or testing of samples seized by police, tended not to be provided in a timely enough manner to inform harm reduction responses and messaging, with services and other stakeholders often relying on *"speculation"* in the event of a cluster of overdoses:

> *A lot of the warnings we get at the moment are based on just speculation. You know, if there is a cluster of deaths then we assume, you get reports that 'oh it's a fentanyl death' with no toxicology, and nothing, you know, to back that up.* (Professional participant 11, NHS)

More timely information was thought to have potential to inform immediate harm reduction responses through a variety of channels. One participant noted that the dissemination of more rapid information about the characteristics of the drug market to people who use drugs could make *"a world of difference"*:

> *You inform people on the same day, you are not going back a week, two weeks, a month, three months later to say 'the Valium on the streets is really strong' and people go 'oh aye I know, I've lost two pals [friends] who have died of it'. You are real time. You are able to go back with that information and say 'here is what people are buying the now'. . . and that will make a world of difference to drug deaths if you are able to do that.* (Professional participant 4, third sector)

Participants noted that DCS could provide a rich picture of drug trends locally, which could enable targeted warnings where substances of concern were in circulation. The capacity to develop a picture of the local market was described as an improvement on existing resources such as WEDINOS:

> *If there was more publicity in a particular area to say 'oh we've had some samples done of drugs in the area at the moment, this is what has come up'. Because with the likes of WEDINOS, that is difficult because you need to know some details about the sample or the reference number to be able to put in. To my knowledge you can't just put in what's been found lately.* (Professional participant 25, NHS)

Participants discussed the information shared from drug checking in two distinct forms. The first was alerts/early warnings shared with a range of parties in the event of a substance of concern being detected. Outreach was often discussed as a means of rapidly disseminating information to communities in the event of a cluster of overdoses. The second form of information in relation to DCS was aggregated drug market monitoring:

> *If there was a drug checking group that met every week and that's shared within a group of maybe 10 people, some of them will be professionals, some of them might be lived experience, you know, peer workers, whatever. . . But very quickly you start to build a kind of information bank of you know 'I started this in October 2021, the cocaine that we were testing in general*

*was 63%. The etizolam was 50%, we were seeing a group of benzodiazepines, in fact we tested eight different benzodiazepines'.* (Professional participant 26, third sector)

Other participants highlighted how drug checking information could help shape service delivery through increasing staff knowledge and enabling services to react to drug market developments more proactively. Relatedly, one participant felt that intelligence gained from drug checking could inform decision making around funding in Alcohol and Drug Partnerships (ADPs; local commissioners of alcohol and drug services in Scotland), for example by highlighting gaps in service provision or emerging trends: *"[drug checking trend information] might tailor what outcomes the local ADPs are making in terms of funding"* (Professional participant 24, NHS).

### Market-level impacts

Participants discussed potential impacts of drug checking at the level of the drug market. They described the potential for increased consumer pressure, as people could choose not to return to a particular supplier or inform their supplier of results. However, caveats were noted to the level of consumer pressure which could be exercised owing to the large and complex nature of the drug market, and the fact that individual choices may be largely dictated by availability and time pressures. The potential for drug sellers to engage in DCS was also discussed.

**Increased pressure from consumers.** Participants noted that drug checking would provide individuals with the opportunity to change their supplier if they found out, through testing, that their drugs were of low quality, had been mis-sold, or contained harmful mixing agents or adulterants. Several participants noted that drug checking would enable them to either switch supplier or return to their supplier and inform them of the composition of a batch:

*I turned up and I have just bought smack [heroin] from somewhere and it's been checked and it's like ten percent purity, you know, I'm not going to go back to that guy.* (PWEDU participant 11)

However, some noted barriers to changing supplier, with one participant stating that drug supplies tend to be inconsistent, constantly changing, and are often disrupted by law enforcement. Another noted that they would likely use drug checking information to change suppliers but may not always have the option to do so. They described how there would likely be occasions where they would be required to return to a supply which was sup-optimal, as sourcing drugs was often labour intensive and limited by availability: *"I'd go to the next door, or I'd go to the next door, and if they haven't got it. . . I will go back and buy them [the original product] anyway"* (PWEDU participant 2).

**Use of drug checking among people who sell drugs.** Participants discussed the potential for people who sell drugs to utilise drug checking as a market level impact of DCS. Those who sell drugs operate at various levels in the supply chain ranging from people selling small quantities to fund their drug use, to those selling large quantities higher in the supply chain, often operating within organised structures [70, 71]. Participants had mixed opinions on whether those selling smaller quantities would use drug checking, with some noting that they may have a lack of interest in checking drugs before selling, due to limited ability to alter the composition of a drug. However, others noted that individuals selling small quantities may use drug checking as "*some of them don't know what they are actually buying*":

*You might get dealers who come in and gae' [give] you it to test it, to see what is actually in the stuff, because some of them just go and buy that half ounce of kit [heroin] and don't actually know what it's been mixed with.* (PWEDU participant 8)

Another participant noted that when they had previously engaged in selling, they would likely have used DCS, as when they sold drugs to someone, they *"[didn't] know if that person is going to die or not"* (PWEDU participant 7). They also described how drug exchanges between individuals were often based on reciprocity and trust, while noting that there were times when this trust was breached:

*If it's somebody that knows me, they will say to me 'oh this is shite or this is good, but I'm just letting you know', ken [you know] some people are really honest and the next person just lies to you.* (PWEDU participant 7)

In relation to those selling larger quantities in a more organised fashion, some participants noted that utilisation of DCS could have substantial harm reduction benefits by providing a form of quality control for larger batches higher up the supply chain. Others expressed scepticism in relation to whether those selling larger quantities or operating at middle market-level would utilise drug checking, feeling that some may be primarily interested in profit and not the wellbeing of people who use drugs. However, it was noted that there may be financial as well as altruistic motivations for using DCS. Some described how those selling could either cut their batch down further after testing, or could use drug checking as verification of a high strength product:

*If the drug dealers found it was like fifty percent or something they'd think 'oh well, we could perhaps cut that down to twenty-five percent'.* (Professional participant 14, NHS)

*People will get the drugs tested and then use that almost, that then becomes their sales pitch. . . I've got the strongest.* (Professional participant 4, third sector)

In relation to DCS being used by suppliers as a form of product verification, some participants felt that this had the potential to increase consumer information on a larger scale than drugs being checked on an individual basis by people who use drugs. However, others felt that this could act to disincentivise drug checking amongst people who use drugs, who would rely on this verification (which may or may not represent the actual product at the point of purchase):

*So, for all you know the first one they gave you was a pure sample, the next one has been cut with talcum powder or baking soda or rat poison, or paracetamol, anything just to make it go further. And that person [who buys the drug] might not get that tested because who wants to use up some of their drugs?* (Professional participant 16, police)

Only one participant discussed the potential for DCS (or peers associated with the service) to actively work with those selling drugs as a means of reducing drug-related harm. They discussed the potential for overdose awareness training with those selling drugs, informing them about drug potency, as well encouraging distribution of naloxone to clients:

*So, the reason I'm saying you need to engage them, and build a relationship with them as well, is because they can become your key players for distributing information, right? Simply by saying 'listen, that heroin you've got is super strong, so it doesn't make sense. . . your client group*

*starts to die of an overdose with it. So, anybody ever talk to you about naloxone, do you know about it? Here, take a couple of supplies of it, just to make sure in case you sell to somebody, and they are just out of prison or something and they overdose and you've got a supply to revive them'.* (Professional participant 4, third sector)

Police participants generally felt that DCS should not engage with those selling drugs. Their view was that drug checking was a harm reduction intervention for those using drugs, not a means of product verification for those selling. Police participants often drew a clear distinction between these two groups, even though research indicates that many people accessing drug checking may engage in both practices [40, 70]. However, one police participant did highlight that it would not be easy to discern between clients using drug checking for individual purposes and those accessing it for the purposes of supply.

**Macro-level impacts.**   Participants discussed potential impacts of DCS at the macro (or population) level. However, some described a need to manage expectations around the impact of DCS and noted these impacts would be challenging to demonstrate.

**Reductions in macro-level indicators of drug-related harm.**   Participants discussed several potential macro-level impacts of drug checking, including reductions in: fatal and non-fatal overdoses; drug-related hospital admissions; drug-related injuries such as injecting injuries; and drug-related deaths. The most cited potential population level impact was a reduction in drug-related deaths. However, participants had varied views on the extent to which DCS would be able to bring about a reduction in deaths at the population level and whether this was a realistic metric on which to judge its success as a harm reduction intervention. Some also asked whether such a complex outcome, driven by numerous factors, could be causally attributed to drug checking. One participant stated that drug checking would need to be able to demonstrate that it had contributed to a reduction in drug-related deaths to be considered a success. However, others were more sceptical about the extent to which drug checking could contribute to a reduction in deaths at the population level, highlighting the need to temper expectations around what DCS could achieve:

> *I don't think the benefits will be. . . it's not like if we do this for a year we are going to stand up and say, you know, 'we've slashed drug deaths' or something like that. I don't think we could do a service like that with that expectation.* (Professional participant 16, police)

One participant noted that drug checking should not be expected to be a *"silver bullet"* for a complex and entrenched public health crisis (Professional participant 26, third sector), and another stated that drug checking could *"contribute a small part"* alongside a wider systemic effort (Professional participant 10, NHS). In relation to drug checking as part of a wider suite of interventions and measures to reduce drug-related deaths, one participant felt that it was the interventions delivered in combination with drug checking which could begin to affect change:

> *[Drug checking alone is] not going to have a huge impact on the drug deaths. But if it's part of a whole range of services then it's yes, you are more likely to be able to save more people's lives.* (Family member participant 2)

Regardless of whether DCS could evidence a population level effect on drug-related deaths, participants with experience of drug use often felt drug checking could *"save a lot of lives"* (PWEDU participant 8), and that it would be worthwhile if it could *"save even one life"* (Family member participant 4). Some participants noted that it would be complex to evidence both

whether drug checking had saved lives, and the extent to which it had contributed to reducing harms at a population level, owing to the myriad of interacting factors which could contribute to both. Untangling the impacts of different services and interventions, often delivered concurrently, was viewed as a particular challenge.

## Discussion

The findings of this paper add to the emerging literature on DCS, highlighting potential impacts across a range of levels from individual service users to wider impacts on drug services, systems, and the drug market [27, 47, 55, 56, 59, 63, 64, 72]. Participants described these meso-, market-, and macro-levels as equally important to the impact of drug checking on individual service users, closely corresponding with the work of Wallace and colleagues on the potential benefits of viewing drug checking through a socio-ecological lens [58]. Participants noted that people at highest risk of drug-related harm may face significant barriers to enacting safer drug use patterns related to stigma, criminalisation, mental health, and the wider risk environment, which is echoed in related literature [73–76]. This suggests that, although individual behavioural outcomes are undoubtedly a vital metric of DCS effectiveness, evaluations of community-based DCS should expand their scope to capture the complex effects of drug checking at multiple levels (micro, meso, macro), and across multiple stakeholders and institutions (individuals, services and systems).

As noted, much of the evidence on drug checking's impact is based on settings serving a particular demographic, described elsewhere as "educated, socially integrated people with high drug literacy who apply several harm reduction strategies" [49, p.7]. Whilst verified or intended disposal of a drug following an unexpected result has been used as a metric of DCS effectiveness in recent evaluations, for community-based services other outcomes capturing more nuanced adaptions in drug use behaviour may be more pertinent, including: using less of a drug or over a longer time period; less mixing of drugs; changing mode of administration; and taking a tester dose (a lower initial amount). Participants in the current study also placed value on broader processes of increased drug literacy and engagement in harm reduction more generally.

Participants discussed a range of potential impacts of DCS at the 'meso-level' (i.e., at the level of services, communities, public health organisations, and formal and informal communication networks). Drug checking was described by frontline staff as a means of building relationships with individuals through the provision of a non-conditional and valued harm reduction intervention. A significant proportion of those accessing DCS are likely to have either never been in contact with health or harm reduction services in relation to their drug use, or to have only engaged in services sporadically [47, 50, 72]. This suggests that DCS have the potential to widen access to other harm reduction provisions including naloxone, sterile injecting equipment, psychosocial support, and medication assisted treatment. DCS were also seen as having potential impacts beyond the services and staff offering the intervention, with potential benefits described for wider frontline services in relation to increasing knowledge of the drug market, facilitating more informed conversations with clients, and helping to identify emerging trends and adapt service provision accordingly in a timely manner. Such outcomes are underexplored in the drug checking literature, suggesting a potential avenue for future evaluations.

The potential for DCS to improve systemic drug market monitoring capacity is well evidenced in the literature [34, 35, 39, 42, 43, 45, 60, 61]. In line with this literature, participants felt that drug checking would improve the currently available drug trend information, providing more rapid information about the drug market. It should be noted that there are challenges

in relation to the capacity of DCS to feed into drug market monitoring systems. Primarily, there should be sufficient coverage in terms of substances tested, with testing sensitive to unknown or novel substances potentially present in low concentrations [30, 61, 77, 78]. Testing that takes place at point-of-care sites can be limited to presumptive results only, due to limitations with equipment. This means that information may not be reliable enough to inform early warnings or public health responses. Building in a fixed laboratory-based component to DCS may, therefore, be important in this regard as it enables confirmatory testing on substances of concern and can validate point-of-care equipment.

Participants discussed the potential impact of DCS at the level of the drug market, echoing emerging findings from North America that drug checking may facilitate forms of grassroots market regulation by providing people who use and sell drugs with reliable information, potentially altering their interactions with the drug market [40, 55, 59, 62, 63]. Previous work has evidenced the use of DCS by people who sell drugs, recording a range of harm reduction practices taken by sellers in response to results. Participants in the current study had mixed views on DCS being used by people who sold drugs. Some noted the positive harm reduction potential of such practices while others, particularly police participants, expressed reservations, including the potential for mis-selling and reducing the perceived need amongst individuals buying drugs to access DCS (as their supplier would have already done so). Existing research has suggested that use of DCS by sellers may reduce perceived need to access the service amongst individuals [62], resulting in diminished engagement with the harm reduction support and advice provided through DCS.

DCS in Scotland, and those responsible for evaluating them, should therefore consider the extent to which they highlight the use of DCS amongst those selling drugs. While evaluation of these practices may evidence harm reduction benefits of DCS extending well beyond individual service users, it may also present challenges relating to the political acceptability and feasibility of DCS. Many DCS are framed, a-politically, as health services, positioning them outside of heated political debates. Evaluation highlighting the value of DCS to those selling drugs may challenge this framing, as well as posing issues regarding licensing compliance for services. However, not attending to the broader potential of DCS may risk silencing debates about the role of the unregulated drug market in producing and exacerbating harms for people who use drugs [58, 59, 70, 79].

The role of DCS in facilitating community action and organisation may be particularly important in contexts facing significant public health crises driven by 'toxic' drug supplies. An example of strategic grassroots organisation is the Drug User Liberation Front (DULF) in Vancouver who use DCS to test drugs in circulation, provide content labels, and redistribute them in an attempt to provide a measure of consumer protection during the ongoing overdose crisis [80]. While these activities are still small-scale, they indicate the potential for DCS to facilitate 'social resistance' [56], market regulation, and communal organising. This highlights how DCS may interact with activity outside of institutional structures [79], for example, how they may intersect with the actions of heroin compassion clubs [58, 81].

Participants discussed the potential for DCS to impact on macro-level indicators of harm such as drug-related hospitalisations and fatal and non-fatal overdose rates. As previously noted, there is currently no evidence that DCS have had an impact on population level indicators of drug-related harm [64], with few evaluations having explored these outcomes. There are likely to be a range of complex methodological challenges associated with such evaluation, including a lack of standardised data, and significant restrictions on access to such data [7]. Aside from methodological challenges, community-based DCS are typically accessed by a relatively small proportion of people who use drugs, which may limit their ability to affect change at a population level [82]. Participants in the current study had mixed views about whether

they thought DCS would have sufficient impact to demonstrate effects on population level indicators of drug-related harm. It was noted that DCS should not be considered as a 'silver bullet', but as one part of a comprehensive harm reduction approach to address the situation in Scotland. This indicates the potential need to manage expectations and carefully communicate the role and potential impact of DCS in Scotland to wider parties including policy makers and the general public.

DCS integrated within overdose prevention sites may hold potential for more concrete evaluation of the impact of DCS on drug-related harms. For example, an evaluation of a DCS integrated into an overdose prevention site in Vancouver, Canada found that, amongst individuals whose substances tested positive (on fentanyl test strips) for fentanyl prior to consumption, 36% intended to reduce their dosage. Further, intended dose reduction was significantly associated with lower odds of overdose on site (OR = 0.41; 95% CI 0.18–0.89) [46]. As consumption takes place on site in overdose prevention sites this may present opportunities for the relationship between DCS engagement and subsequent drug use outcomes to be more clearly linked and evaluated.

## Implications for policy, practice and research

While DCS, and the drug market trend information they produce and disseminate, may have value beyond individuals accessing the service, there has been no systematic evaluations of the impact of drug checking across these multiple levels [27, 64]. This underlines the potential for mixed-methods evaluation focusing on multiple aspects of DCS delivery and outcomes. In addition to more measurable outcomes such as intended behaviour change, evaluations should focus on less tangible impacts requiring qualitative methods such as interviews, ethnography, and case studies. Additionally, widening the scope of evaluation to consider how drug checking information is used by wider organisations, and its interaction with public health and harm reduction structures, may be important to capturing the role of DCS more broadly.

DCS have often operated without sufficiently funded and robust evaluations, sustaining gaps in the evidence base. It should also be noted that, while a number of potential outcomes of DCS can be identified in theory, the impact of DCS in practice will depend on several factors including: location; service user demographics and type of drug use; model of service delivery; funding; equipment and methods utilised; and a range of further logistical considerations [33, 61, 83–88]. For example, the capacity for DCS trend information to reach and impact on a wide range of organisations and individuals depends on the development of effective communication structures, potentially requiring significant time and resource investment. Additionally, system-level impacts may not be feasible for small-scale services accessed by limited numbers of individuals. It is important for evaluations to explore technical, political, capacity, and resource challenges which may limit the effectiveness of DCS in order to foster ongoing innovation.

## Strengths and limitations

This paper builds on existing research concerning the potential impacts of drug checking [58] and suggests some initial avenues for future research and evaluation. A contribution of the paper is the suggestion that evaluations look for impacts outside of drug checking sites by expanding the scope of inquiry to wider services and public health structures, as well as networks of people who use and sell drugs, using the micro, meso, and macro framing of the impact of DCS outlined by Measham (2019, 2020) [27, 47]. Whilst recognising significant challenges to pursuing these lines of inquiry, widening the scope of exploration holds potential for building a richer picture of the impacts of community-based DCS.

This paper has limitations to consider. Interviews were conducted pre-implementation amongst a group of stakeholders with no experience of delivering DCS. Consistent with much pre-implementation research, the findings represent the impacts that people hope or feel drug checking may have, rather than concrete descriptions of actual impacts. Accordingly, the paper's findings may be read as naïve by some, in that they may be perceived as presenting a range of potential harm reduction outcomes without due consideration of the challenges and limitations faced by DCS in practice. To address this limitation, and as a means of grounding the research in the practical constraints faced by DCS, the discussion section has outlined some of these challenges.

## Conclusion

DCS are complex interventions which may have impacts on individuals using the service as well as on broader public health systems and processes. This paper has presented some potential lines of inquiry for future evaluations in Scotland and elsewhere. There is a need for well-funded, multi-method evaluations to rigorously assess the potentially wide-ranging impact of DCS. It is important that community-based DCS are not viewed solely as individual-level harm reduction interventions in the context of complex and rapidly evolving illicit drug markets and increasing levels of harm.

## Acknowledgments

We would like to thank wider members of the research/project team for their invaluable input through the study including: Saket Priyadarshi, Jo McManus, Fiona Raeburn, Laura Rothney, Simon Rayner, Kirsty Licence, Emma Fletcher, and Emma Crawshaw. We would like to acknowledge members of the Lived Experience Reference Group: Mike Hunter, Sam Raoin, Victoria Grover, Lee Caldwell and Phil Foley. We would also like to thank all members of the wider Project Advisory Group for providing their time and expertise throughout the study. Finally, we would like to extend our appreciation to all the participants for their time and their willingness to be involved.

## Author Contributions

**Conceptualization:** Tessa Parkes, Hannah Carver, Elizabeth V. Aston.

**Data curation:** Danilo Falzon, Wendy Masterton.

**Formal analysis:** Danilo Falzon, Tessa Parkes, Hannah Carver, Wendy Masterton.

**Funding acquisition:** Tessa Parkes, Hannah Carver, Elizabeth V. Aston.

**Investigation:** Tessa Parkes, Hannah Carver, Vicki Craik, Elizabeth V. Aston.

**Methodology:** Danilo Falzon, Tessa Parkes, Hannah Carver, Wendy Masterton, Bruce Wallace.

**Writing – original draft:** Danilo Falzon, Tessa Parkes, Hannah Carver, Wendy Masterton.

**Writing – review & editing:** Tessa Parkes, Hannah Carver, Wendy Masterton, Bruce Wallace, Vicki Craik, Fiona Measham, Harry Sumnall, Rosalind Gittins, Carole Hunter, Kira Watson, John D. Mooney, Elizabeth V. Aston.

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
