## [Decision Letter · Decision Letter 0]

25 Aug 2023

PONE-D-23-18649“It would really support the wider harm reduction agenda across the board”: A qualitative study of the potential impacts of drug checking service delivery in Scotland.PLOS ONE

Dear Dr. Falzon,

Thank you for submitting your manuscript to PLOS ONE. After careful consideration, we feel that it has merit but does not fully meet PLOS ONE’s publication criteria as it currently stands. Therefore, we invite you to submit a revised version of the manuscript that addresses the points raised during the review process.

We look forward to receiving your revised manuscript.

Kind regards,

Meghana Ray, Ph.D., MBA, B.Pharm

Academic Editor

PLOS ONE

Reviewers' comments:

Reviewer's Responses to Questions

**Comments to the Author**

1. Is the manuscript technically sound, and do the data support the conclusions?

Reviewer #1: Yes

Reviewer #2: Yes

2. Has the statistical analysis been performed appropriately and rigorously? 

Reviewer #1: Yes

Reviewer #2: N/A

3. Have the authors made all data underlying the findings in their manuscript fully available?

Reviewer #1: No

Reviewer #2: No

4. Is the manuscript presented in an intelligible fashion and written in standard English?

Reviewer #1: Yes

Reviewer #2: Yes

5. Review Comments to the Author

Reviewer #1: Excellent qualitative study looking at the potential benefits of implementation of a drug checking service in Scotland from relevant stakeholders.

Here are a couple minor edits:

- Line 287: Would recommend changing "drug consumption" to "drug composition"

- Line 336: Change "on" to "or"

Reviewer #2: Thank you for the opportunity to comment on this paper. The paper is extremely well-written and the introductory section in particular is quite informative, the authors nicely synergize the existing literature on drug checking services and lay out the purpose of the current qualitative examination of potential DCS impacts. The manuscript is timely given the ongoing urgency to reduce overdose deaths and the increasing complexities of and adulterants in the drug supply, as well as the challenges in improving engagement with harm reduction services and extending the reach of services to more people who would benefit. The qualitative sample is of sufficient size and the coding methods are well-described, which supports the analyses with good rigor. A deeper understanding of potential multi-level impacts of DCS may inform strategies to optimize their implementation. The paper does a good job laying out future research and evaluation needs for DCS. In sum, this paper has potential to make an important contribution to the literature.

In my view the primary weakness of the paper relates to the somewhat constrained presentation of findings; the authors note that the interview guide asked about a wide range of topics,

including: challenges around implementation; the parties who should be involved in implementation; suitable locations and service delivery models; barriers and facilitators to

engagement amongst different groups of people who use drugs; policing and community

challenges to delivery; general perceptions of drug checking; and the potential impacts

of delivering DCS. This set an expectation for the reader that we would see some integration of data from these different, intertwined domains, yet we see only the isolated findings on potential impact. Understanding it is challenging to include the wealth of qualitative information garnered on these other issues, the authors may consider more clearly laying out the paper’s scope.

As a minor point, some clarification in the methods may be useful, specifically in the description of coding versus the thematic analysis. The authors state that “Initial coding was conducted using a thematic analysis approach [68].” This is somewhat difficult to reconcile, perhaps the intention is to state that the initial coding was conducted using an inductive approach? This should be clarified. I also noted that the overarching “themes” identified in the results table simply seem to be areas of inquiry from the interview guide, they do not appear to be emergent or developed themes, whereas the “subthemes” actually seem to be the novel emergent thematic content yielded by the analytic process. The authors should consider adjusting their labelling and terminology accordingly.

6. PLOS authors have the option to publish the peer review history of their article (what does this mean?). If published, this will include your full peer review and any attached files.

Reviewer #1: **Yes: **Mark Lysyshyn

Reviewer #2: No

---

## [Author Response · Author response to Decision Letter 0]

30 Aug 2023

Please see the separate document uploaded called 'Response to Reviewers' which contains the responses in a suitably formatted manner.

EDITOR'S COMMENTS

Editor comment 1:

Response to editor comment 1:

Thank you for pointing this out. 

We have changed the file names (for both the original manuscript and the document with tracked changes) to reflect PLOS ONE’s requirements. We have also changed the size of headings in the paper to meet the requirements outlined in the templates provided (size 18 for main headings, 16 for sub-headings and 14 for third headings). 

We have changed the line spacing to double as specified in the template and have also added page numbers to the manuscript.

As far as we can see, these are the areas where the manuscript did not align with PLOS ONE’s style requirements but please let us know if there is something we have missed, and we can amend accordingly. 

Editor comment 2:

We note that the grant information you provided in the ‘Funding Information’ and ‘Financial Disclosure’ sections do not match.

Response to editor comment 2:

Thank you. We have now changed the information provided in the funding information section to: ‘Corra Foundation/Drug Deaths Taskforce’ grant number 20/5304, which corresponds with the information provided in the financial disclosure section.

Editor comment 3: 

In your Data Availability statement, you have not specified where the minimal data set underlying the results described in your manuscript can be found. PLOS defines a study's minimal data set as the underlying data used to reach the conclusions drawn in the manuscript and any additional data required to replicate the reported study findings in their entirety. All PLOS journals require that the minimal data set be made fully available. For more information about our data policy, please see http://journals.plos.org/plosone/s/data-availability.

Upon re-submitting your revised manuscript, please upload your study’s minimal underlying data set as either Supporting Information files or to a stable, public repository and include the relevant URLs, DOIs, or accession numbers within your revised cover letter. For a list of acceptable repositories, please see http://journals.plos.org/plosone/s/data-availability#loc-recommended-repositories. 

Any potentially identifying patient information must be fully anonymized.

Response to editor comment 3: 

Thank you. We are not able to make transcripts publicly available and would have ethical concerns about doing so. Given that recruitment for the study took place through a relatively small number of organisations in each city, it would not be possible to guarantee participant anonymity. Additionally, we did not ask for participants consent to make the transcripts publicly available given the potentially sensitive nature of the topics being discussed (e.g., drug use and buying practices, which are criminalised, amongst participants who use drugs).

We have provided the following data availability statement: 

‘We have significant ethical concerns about making the transcripts available in a public repository. The small number of participants in each city makes it impossible to guarantee that participants cannot be identified through the transcripts. We also did not ask for our participants' consent to make the transcripts available publicly. This was a choice based on the sensitive nature of the topics covered in the interviews including criminalised activity. Other researchers who want to replicate the study can contact the corresponding author and/or the NICR ethics committee at University of Stirling (ethics@stir.ac.uk).’ 

In providing this information, we are following the guidance outlined on PLOS ONE’s ‘data access restrictions’ page where it outlines that if there are ethical or legal restrictions to sharing a sensitive data set, authors should explain these restrictions (e.g., potentially sensitive and identifiable information) and should also provide a relevant institutional body (e.g., ethics committee) to which data requests may be sent, which we have now stated in our data availability statement. 

Editor comment 4: 

Please review your reference list to ensure that it is complete and correct. If you have cited papers that have been retracted, please include the rationale for doing so in the manuscript text or remove these references and replace them with relevant current references. Any changes to the reference list should be mentioned in the rebuttal letter that accompanies your revised manuscript. If you need to cite a retracted article, indicate the article’s retracted status in the References list and also include a citation and full reference for the retraction notice.

Response to editor comment 4: 

Thank you. We have checked every reference and made some small changes to ensure that they are correct and complete. These are primarily small grammatical/formatting corrections, or including hyperlinks for DOIs where missing. 

One reference was changed:

Ref 71: When checking the references, we found that the link for ref 71 no longer worked and we could not find it online despite multiple searches. We therefore changed the reference. However, the new reference is by same author and reflects their body of work and many of the same points as the original reference. It is appropriate and suitable to substantiate the points being made in our paper. 

We hope that these changes mean that the references are now aligned with PLOS ONE’s criteria. 

REVIEWER ONE COMMENTS:

Reviewer one comment 1: 

Excellent qualitative study looking at the potential benefits of implementation of a drug checking service in Scotland from relevant stakeholders.

Response to reviewer one comment 1:

Thank you for reviewing the paper and for your supportive comments. 

Reviewer one comment 2:

Line 287: Would recommend changing "drug consumption" to "drug composition"

Response to reviewer one comment 2:

We have made this change: the sub-heading now reads ‘increased availability of information about drug composition’.

This change can be found on line 298, pg.13.

Reviewer one comment 3:

Line 336: Change "on" to "or"

Response to reviewer one comment 3:

Thank you, we have changed this to ‘or’: 

‘Participants tended to place emphasis on potential changes to patterns of drug use, rather than on disposal of substances in the event of an unexpected or concerning result’. 

This change can be found on line 355, pg.15

REVIEWER TWO COMMENTS:

Reviewer two comment 1:

Thank you for the opportunity to comment on this paper. The paper is extremely well-written and the introductory section in particular is quite informative, the authors nicely synergize the existing literature on drug checking services and lay out the purpose of the current qualitative examination of potential DCS impacts. The manuscript is timely given the ongoing urgency to reduce overdose deaths and the increasing complexities of and adulterants in the drug supply, as well as the challenges in improving engagement with harm reduction services and extending the reach of services to more people who would benefit. The qualitative sample is of sufficient size and the coding methods are well-described, which supports the analyses with good rigor. A deeper understanding of potential multi-level impacts of DCS may inform strategies to optimize their implementation. The paper does a good job laying out future research and evaluation needs for DCS. In sum, this paper has potential to make an important contribution to the literature.

Response to reviewer two comment 1:

Thank you for these supportive comments outlining the paper strengths. We have taken a number of steps to address the constructive and helpful comments you make (see below).

Reviewer two comment 2:

In my view the primary weakness of the paper relates to the somewhat constrained presentation of findings; the authors note that the interview guide asked about a wide range of topics, including: challenges around implementation; the parties who should be involved in implementation; suitable locations and service delivery models; barriers and facilitators to engagement amongst different groups of people who use drugs; policing and community challenges to delivery; general perceptions of drug checking; and the potential impacts of delivering DCS. This set an expectation for the reader that we would see some integration of data from these different, intertwined domains, yet we see only the isolated findings on potential impact. Understanding it is challenging to include the wealth of qualitative information garnered on these other issues, the authors may consider more clearly laying out the paper’s scope.

Response to reviewer two comment 2:

Thank you for this helpful comment. We appreciate that having to limit the scope of the paper omits a lot of interesting data on issues such as implementation. Other papers from the project have tackled these topics, so a decision was made for this paper to focus only on potential harm reduction impacts. We would also like to note the absence of literature on drug checking impacts which is why we specifically chose to write this paper as a separate contribution. 

That said, we appreciate that the way in which we discuss this in the methods section may be confusing and lead the reader to believe that the paper will focus on the wider considerations mentioned. The section which the reviewer points to as problematic is as follows:

‘Interview schedules were designed to ask about a wide range of issues relating to drug checking, including: challenges around implementation; the parties who should be involved in implementation; suitable locations and service delivery models; barriers and facilitators to engagement amongst different groups of people who use drugs; policing and community challenges to delivery [23]; general perceptions of drug checking; and the potential impacts of delivering DCS.’

In order to address the comments made, we have added the following, leading directly on from the above quote:

‘Participant discussion of the potential harm reduction impact of DCS is the focus of the current paper. Other papers from the study explore topics such as policing of DCS [23], and key implementation and service design considerations [66, 67].’

This addition can be found on lines 241-244, pg. 10-11. 

We hope that this helps make the framing of the current paper clearer and signposts the reader to our other published work from this study tackling issues such as service design and implementation. 

We should be clear that the data being presented in this paper is completely original and there is no duplication between the papers we note above and the one submitted here to PLOS ONE. We have also ensured that the background and discussion sections are unique contributions. 

Reviewer two comment 3:

As a minor point, some clarification in the methods may be useful, specifically in the description of coding versus the thematic analysis. The authors state that “Initial coding was conducted using a thematic analysis approach [68].” This is somewhat difficult to reconcile, perhaps the intention is to state that the initial coding was conducted using an inductive approach? This should be clarified.

Response to reviewer two comment 3:

Thank you for pointing this out. We have made a slight change to clarify this point, and the text now reads: 

‘Data were analysed using thematic analysis [68]. Initial coding was conducted using an inductive approach, with a selection of transcripts (n=16) from a mix of participant groups coded by DF to create an initial coding framework.’ (Lines 246-249, pg.11). 

Reviewer two comment 4:

I also noted that the overarching “themes” identified in the results table simply seem to be areas of inquiry from the interview guide, they do not appear to be emergent or developed themes, whereas the “subthemes” actually seem to be the novel emergent thematic content yielded by the analytic process. The authors should consider adjusting their labelling and terminology accordingly.

Response to reviewer two comment 4:

Thank you for this helpful observation. We have made changes to the labelling and terminology. We now state that: 

‘Themes are reported under four overarching “levels” of potential impact: individual-level impacts; meso-level impacts; market-level impacts; and macro-level impacts’ (Lines 280-282, pg.12)

We have also changed the table showing the themes. We have removed the text describing the ‘levels’ of influence (individual, meso, market, macro) as ‘themes’. We have re-labelled what were previously ‘sub-themes’ as ‘themes’. We now have seven themes deductively organised under four levels of influence. (Table 1, pg. 12-13)

We believe that this addresses the reviewer’s important comment.

---

## [Editor Report · Decision Letter 1]

28 Sep 2023

“It would really support the wider harm reduction agenda across the board”: A qualitative study of the potential impacts of drug checking service delivery in Scotland.

PONE-D-23-18649R1

Dear Dr. Falzon,

We’re pleased to inform you that your manuscript has been judged scientifically suitable for publication and will be formally accepted for publication once it meets all outstanding technical requirements.

Kind regards,

Meghana Ray, Ph.D., MBA, B.Pharm

Academic Editor

PLOS ONE
---

## [Editor Report · Acceptance letter]

10 Oct 2023

PONE-D-23-18649R1 

*“It would really support the wider harm reduction agenda across the board”*: A qualitative study of the potential impacts of drug checking service delivery in Scotland. 

Dear Dr. Falzon:

I'm pleased to inform you that your manuscript has been deemed suitable for publication in PLOS ONE. Congratulations! Your manuscript is now with our production department. 

Kind regards, 

on behalf of

Dr. Meghana Ray 

Academic Editor

PLOS ONE